# Real-World Efficacy and Adherence to Palbociclib in HR-Positive, HER2-Negative Advanced Breast Cancer: Insights from a Romanian Cohort

**DOI:** 10.3390/cancers16244161

**Published:** 2024-12-13

**Authors:** Cristian Virgil Lungulescu, Georgiana-Cristiana Camen, Mihaela-Simona Naidin, Tradian-Ciprian Berisha, Andrei Bita, Venera-Cristina Dinescu, Sandra Alice Buteica, Marina-Daniela Dimulescu, Simona Ruxandra Volovat, Adina Turcu-Stiolica

**Affiliations:** 1Oncology Department, University of Medicine and Pharmacy of Craiova, 200349 Craiova, Romania; cristian.lungulescu@umfcv.ro; 2Radiotherapy Department, University of Medicine and Pharmacy of Craiova, 200349 Craiova, Romania; 3Pharmaceutical Management and Marketing Department, University of Medicine and Pharmacy of Craiova, 200349 Craiova, Romania; mihaela.subtirelu@umfcv.ro (M.-S.N.); adina.turcu@umfcv.ro (A.T.-S.); 4Doctoral School, University of Medicine and Pharmacy of Craiova, 200349 Craiova, Romania; berishaciprian@gmail.com (T.-C.B.); marina.dimulescu@umfcv.ro (M.-D.D.); 5Pharmacognosy Department, University of Medicine and Pharmacy of Craiova, 200349 Craiova, Romania; andrei.bita@umfcv.ro; 6Department 6—Health Promotion and Occupational Medicine, University of Medicine and Pharmacy of Craiova, 200349 Craiova, Romania; venera.dinescu@umfcv.ro; 7Pharmaceutical Chemistry Department, University of Medicine and Pharmacy of Craiova, 200349 Craiova, Romania; alice.buteica@umfcv.ro; 8Department of Medical Oncology, University of Medicine and Pharmacy Grigore T. Popa Iasi, 700115 Iasi, Romania; simonavolovat@gmail.com

**Keywords:** palbociclib, overall survival, time to treatment discontinuation, adherence

## Abstract

In Romania, around 12,000 new cases of breast cancer are diagnosed each year, making it the second most common cause of cancer-related deaths, after lung cancer. The objective of this study is to evaluate the efficacy, safety, and adherence to Palbociclib in combination with either an aromatase inhibitor or fulvestrant in a real-world cohort of HR+/HER2− breast cancer patients from Romania. By examining clinical outcomes, such as time to treatment discontinuation, overall survival, and adherence, this study aims to contribute to the growing body of real-world data that support the use of Palbo in routine clinical practice. Understanding these factors in a real-world setting is essential for optimizing treatment strategies and improving patient outcomes globally, particularly in underrepresented regions where trial data are scarce. Our findings confirm pre-existing trial and real-world data, showing no significant differences in overall survival between the two treatment options.

## 1. Introduction

Globally, breast cancer remains the most frequently diagnosed malignancy among women and a leading cause of cancer-related mortality. It accounts for nearly 25% of all cancer cases in women, with over two million new cases diagnosed annually [1]. In Romania, around 12,000 new cases of breast cancer are diagnosed each year, making it the second most common cause of cancer-related deaths, following lung cancer [2]. Despite advancements in early detection and treatment, the burden of this disease continues to grow, necessitating effective therapeutic strategies to improve outcomes. According to Kardas et al., medication adherence is a key enabler of optimal health outcomes [3]. For patients with hormone receptor-positive, HER2-negative (HR+/HER2−) advanced breast cancer, the combination of cyclin-dependent kinase 4/6 inhibitors (CDK4/6i) with endocrine therapies has revolutionized treatment. Palbociclib (Palbo), the first CDK4/6i approved for this indication, has demonstrated significant clinical benefits in landmark trials such as PALOMA-1 and PALOMA-3 [4,5]. 

However, the translation of clinical trial findings into real-world settings presents unique challenges. Clinical trials often exclude patients with comorbidities or advanced age, limiting the generalizability of their results. Palbo adherence presents a challenge to breast cancer management, as patients receive 125 mg of Palbo orally once daily from day 1 to day 21 of every 28-day cycle, followed by 7 days off of the treatment. Additionally, the combination of Palbo with an aromatase inhibitor (AI) (orally once-daily continuously) or fulvestrant (intramuscularly on days 1 and 14 of cycle 1, every 28 days (±7 days) thereafter starting) makes it difficult to comply with the treatment. Some studies reported an association between Palbo non-adherence and factors such as income (less than $75,000) [6] and follow-up duration [7]. Furthermore, real-world data highlight additional barriers, including medication adherence, treatment tolerability, and access to therapies, especially in resource-limited settings [6]. Very little is known about the association between Palbo adherence and overall survival (OS) [8]. Although the oral route is preferred owing to several advantages, including flexible dosage options, ease of administration, and a reduced risk of infection and contamination, previous studies reported negative health consequences of oral anti-cancer medication non-adherence in breast cancer, but only for adjuvant endocrine therapy [9]. There were statistically significant positive associations identified between endocrine therapy non-adherence and the risk of distant metastasis in patients with BC, recurrence of BC, worse disease-free survival, or mortality [8].

Cyclin-dependent kinases 4 and 6 (CDK4/6) are pivotal in regulating the cell cycle, driving the transition from the G1 to the S phase. In hormone receptor-positive (HR+), human epidermal growth factor receptor 2-negative (HER2-) breast cancer, the dysregulation of the CDK4/6 pathway leads to uncontrolled cellular proliferation, which is central to disease progression. CDK4/6i, including Palbo, ribociclib, and abemaciclib, have revolutionized the treatment landscape for HR+, HER2− advanced breast cancer. These inhibitors, particularly Palbo, halt cell cycle progression, inducing cellular senescence and apoptosis, making them a cornerstone in managing metastatic breast cancer [3,10].

Palbo, in combination with endocrine therapy (ET), such as AI or fulvestrant, has demonstrated significant efficacy in prolonging progression-free survival (PFS) and overall survival (OS) in multiple clinical trials. The PALOMA trials, which assessed Palbo’s effectiveness in HR+/HER2− breast cancer, set the foundation for its approval as both a first-line and second-line therapy. PALOMA-1 showed a remarkable improvement in PFS when Palbo was added to letrozole compared to letrozole alone, while PALOMA-3 extended these findings to patients who had progressed on prior endocrine therapy, demonstrating its effectiveness in a broader range of patients [4,11].

Despite these successes in clinical trials, real-world data (RWD) are essential for understanding the performance of Palbo in diverse patient populations, particularly in settings outside the rigor of controlled trials. Adherence, side effects, and patient management play crucial roles in determining real-world outcomes, as demonstrated in studies showing high adherence to Palbo in clinical practice, yet highlighting factors such as toxicity management that influence long-term use. These challenges underscore the importance of understanding the performance of Palbo in routine clinical practice, particularly in underrepresented populations such as those in Romania.

This study aims to address this gap by evaluating the real-world efficacy, safety, and adherence to Palbo in combination with AIs or fulvestrant among HR+/HER2− advanced breast cancer patients in Romania. By analyzing outcomes such as time to treatment discontinuation (TTD), overall survival (OS), and adherence, this research contributes to the growing body of real-world evidence. It provides valuable insights into the use of Palbo in routine clinical settings, helping to optimize treatment strategies and improve patient outcomes in a region where such data are scarce.

## 2. Materials and Methods

### 2.1. Study Population

Our retrospective study included patients with breast cancer who were receiving Palbo. Data were extracted from the electronic database of the Romanian Health Insurance House, Dolj County, for the disease code 124 (the code for breast cancer in our health insurance system). Ethical approval for our research, obtained under Ethics Council approval number 175/29.10.2021, permitted us to access anonymized patient data from community pharmacies in Dolj County, Romania, as reported to the Dolj Health Insurance House. This study focused on data spanning the past six years, from 2018 to 2023, with the first reimbursed prescription for Palbo issued in July 2018. We excluded patients who began Palbo treatment in 2022 or 2023 to ensure a follow-up period of more than two years. Additionally, we excluded patients that did not receive Palbo treatment for at least two months. Available variables included age, gender, death date, start and end of treatment, type of treatment, and follow-up data. No information about the reason for discontinuation of treatment was available.

### 2.2. Outcomes

The primary outcome was OS, with secondary outcomes including time to TTD and adherence to Palbo. OS was defined as the time in months from the initiation of Palbo treatment to death from any cause. If patients did not die, they were censored at the study cutoff date (31 December 2023). TTD was defined as the duration in months from the initiation of Palbo treatment to its termination (no reimbursed prescription). Adherence to Palbo was assessed using prescription refill data from electronic records, calculated as the medication possession ratio (MPR). MPR was determined by the ratio of the ∑months of medication supply to the ∑months between the first and the last month of prescription. MPR ranges between 0 and 1, with a value greater than 0.8 indicating that the patient was adherent to the treatment. Switching between different CDK 4/6i within the treatment line was not considered in the adherence calculation.

### 2.3. Statistical Analysis

We used R packages (R Core Team 2022, v. 4.2.2 for Windows) [12,13,14], for the statistical analysis. Descriptive statistics were performed, with continuous variables presented as the mean ± standard deviation and median (interquartile range), while discrete variables were presented as frequencies and percentages. The normality of continuous variables was assessed using the Shapiro–Wilk test, where a *p*-value less than or equal to 0.05 indicated a rejection of the null hypothesis of normality. Kruskal–Wallis testing was conducted to compare Palbo adherence levels across patient groups, with violin-plots provided for visual interpretation. The heatmap correlation matrix included Spearman’s correlation coefficients, with green representing a positive correlation (Spearman’s ρ = 1) and orange representing a negative correlation (Spearman’s ρ = −1). The Kaplan–Meier method was used to estimate TTD and OS with 95% confidence intervals (CIs). The log-rank test was used to compare groups by treatment or by initiation year of Palbo treatment. We performed an unadjusted analysis (without controlling baseline patient characteristics or any sensitivity analysis). The Cox proportional hazards regression model was applied to estimate the hazard ratio (HR) with a corresponding 95%CI for assessing the influence of factors such as age, gender, and adherence on OS or TTD. A *p*-value less than 0.05 was considered statistically significant.

## 3. Results

### 3.1. Study Population

During 2018 and 2023, a total of 224 patients initiated treatment with Palbo plus endocrine therapy as first-line metastatic. We excluded 105 patients, due to missing mortality data (*n* = 3), short follow-up periods (*n* = 84), or receiving less than two months (with one or two months) of Palbo treatment (*n* = 12). The final cohort consisted of 125 patients receiving Palbo and endocrine treatment (female: 123, 98%; mean age ± SD: 61.9 ± 12.1 years; median (IQR) age: 62 (53, 70) years). Two combinations with endocrine treatment were observed: Palbo + AI (104, 83%), and Palbo + fulvestrant (21, 17%).

### 3.2. Time to Treatment Discontinuation and Overall Survival

The median TTD was 19 months (95%CI, 19.3–24.9 months) (Figure 1A). Among 125 patients, 46 (37%) died before data cut-off (37 patients treated with Palbo + AI, 9 patients treated with Palbo + fulvestrant). The median OS was 39 months (95%CI, 33.8–40.1 months) (Figure 1B).

### 3.3. Comparation of Time to Treatment Discontinuation and Overall Survival Between the Palbo + AI Group and Palbo + Fulvestrant Group

When treatment involved Palbo + AI, the median TTD was not available [NA] (95%CI, 27.0-NR) months. We observed that when treatment involves Palbo + AI, the survival curve does not drop below ½ during the observation period, thus the survival interval is undefined (Figure 2A). When treatment involved Palbo + fulvestrant, the median TTD was 25.0 (95%CI, 13.0-NA) months. After performing Cox regression, we observed that when treatment involves Palbo + fulvestrant, there is a 1.53 (0.65–2.80, *p* = 0.42)-times risk than when treatment involves Palbo + AI. Related to TTD, no significant differences were found between the two types of treatments (χ^2^ = 1.33, df = 1, log-rank *p* = 0.249).

When treatment involved Palbo + AI, the median OS was NA (95%CI, 54.8-NA) months. We observed that when treatment involves Palbo + AI, the survival curve does not drop below 1/2 during the observation period, thus the survival interval is undefined (Figure 2B). When treatment involved Palbo + fulvestrant, the median OS is 50.8 (95%CI, 34.1-NA) months. After performing Cox regression, we observed that when treatment involves Palbo + fulvestrant, there is a 1.19 (0.57–2.48, *p* = 0.638)-times risk than when treatment involves Palbo + AI. Related to OS, no significant differences were found between the two types of treatments (χ^2^ = 0.221, df = 1, log-rank *p* = 0.638). 

The 12- and 36-month TTD rates were higher for Palbo combined with AI than combined with fulvestrant, as is summarized in Table 1.

The 24- and 36-month OS rates were higher for Palbo combined with AI than combined with fulvestrant, as is summarized in Table 2.

### 3.4. Comparation of Time to Treatment Discontinuation and Overall Survival Between the Two Lines of Treatment, Hormone Therapy-Naïve (HTN) and Prior Adjuvant Therapy (PAT)

Patients in HTN treatment with Palbo had a significantly higher TTD than patients in PAT (Log-rank *p* = 0.0036), as shown in Figure 3A. The median TTD in HTN is NA [95%CI, 40-NA] months. We note that the survival curve for TTD in HTN patients does not drop below 1/2 during the observational period, thus the median survival is undefined. For patients in PAT, the median TTD is 20 [95%CI, 10-NA] months. Applying Cox regression, there is a 2.64 (1.34–5.18, *p* = 0.005)-times greater risk for patients in PAT than patients in HTN. 

The combination of first-line Palbo HTN therapy with endocrine therapy had shown a significant improvement in OS over the combination of second-line Palbo with endocrine therapy (Log-rank *p* = 0.0094), as shown in Figure 3B. The median OS in HTN is NA [95%CI, NA-NA] months. We note that the survival curve for OS in HTN patients does not drop below 1/2 during the observational period, thus the median survival is undefined. For patients in PAT, median OS is 32.9 [95%CI, 21.9-NA] months. Applying Cox regression, we demonstrated significantly longer OS in patients receiving Palbo in HTN compared with PAT (hazard ratio HR, 2.37; 95%CI, 1.21–4.62; *p* = 0.012).

The TTD rates at 12, 24, 36, and 60 months were higher in HTN combination than in PAT combination with Palbo, as shown in Table 3.

The OS rates at 24, 36, and 60 months were higher in HTN combination than in PAT combination with Palbo, as shown in Table 4.

#### 3.4.1. Multivariable Analysis Assessing the Influence of Factors Such as Treatment Type, Age, Gender, and Adherence on OS or TTD 

The effect of treatment type, age, adherence, or gender on OS was not statistically significant, as in Table 5.

In our analysis, treatment type, age, adherence, and gender were not significantly associated with TTD, as in Table 6.

#### 3.4.2. Comparation of Time to Treatment Discontinuation and Overall Survival Between Females and Males

For females, the 12-, 24-, 36-, and 60-month TTD was 76.5% [95%CI, 68.9–84.9%], 61.1% [95%CI, 52.2–71.6%], 56.3% [95%CI, 46.8–67.7%], and 50.9% [95%CI, 40.4–64.2%], respectively. For males, the 12-month TTD and 24-month TTD were 100% [95%CI, 100–100%]. No differences were observed among females and males regarding the TTD of Palbo therapy (log-rank test, χ^2^ = 0.0661, df = 1, *p* = 0.797).

For females, the 12-, 24-, 36-, and 60-month OS was 87.8% [95%CI, 82.2–93.8%], 77.2% [95%CI, 70.2–85.0%], 67.8% [95%CI, 59.8–76.9%], and 57.5% [95%CI, 47.9–68.9%], respectively. For males, the 12-, 24-, and 36-month OS was 100% [95%CI, 100–100%], 100% [100–100%], and 50% [95%CI, 12.5–100%], respectively. No differences were observed among females and males regarding the OS of Palbo therapy (log-rank test, χ^2^ = 0.0497, df = 1, *p* = 0.824).

### 3.5. Palbociclib Adherence Analysis

The mean ± SD adherence in our study was 0.91 ± 0.1 (range, 0.33–1) with a proportion of 14% of non-adherent patients taking Palbo for a 0.8 adherence cut-off. We observed no differences among the Palbo adherence according to the year of inclusion (*p* = 0.13), with lower adherence in 2018 (mean ± SD, 0.85 ± 0.18; median (IQR), 0.93 (0.7–1.0)) than in 2019 (mean ± SD, 0.88 ± 0.18; median (IQR), 1 (0.77–1.0)), 2020 (mean ± SD, 0.95 ± 0.09; median (IQR), 1.0 (0.93–1.0)), and 2021 (mean ± SD, 0.94 ± 0.1; median (IQR), 1.0 (0.94–1.0)), as shown in Figure 4.

No correlations were found between adherence to Palbo and OS or TTD (ρ_adherence and OS_ = 0.04, *p* = 0.593; ρ_adherence and TTD_ = −0.07, *p* = 0.982), as shown in Figure 5.

## 4. Discussion

Despite the ongoing research and development of novel cancer therapies, including cutting-edge pharmaceutical forms like engineered nanoparticles for targeted drug delivery [15], and the study of natural compounds for their potential in cancer treatment [16], the landscape of cancer care is still based on conventional therapy, such as the use of the CDK4/6i Palbo in the treatment of breast cancer.

Endocrine treatment is a cornerstone of first-line treatment for advanced breast cancer with HR-positive/HER2-negative disease. The introduction of CDK 4/6i has significantly improved both progression-free and overall survival in these patients compared to endocrine treatment alone, currently being the standard of care in first-line settings. 

Real-world data provide additional insights into clinical trials by providing insights into treatment tolerability, dose modifications, and efficacy in routine clinical practice. Unlike randomized clinical trials, which are restricted by strict inclusion and exclusion criteria, real-world studies are more reflective of the broader patient population encountered in everyday practice, making their findings more generalizable. 

This study reports the first Romanian real-word data for Palbo, the first CDK 4/6i approved for this setting, in combination with endocrine treatments such as aromatase inhibitors and fulvestrant. Palbo was approved in Romania in 2018 in both first- and second-line settings for metastatic breast cancer. 

In the current analysis, we report data on 125 patients diagnosed with HR-positive/HER2-negative metastatic breast cancer who were prescribed Palbo in combination with endocrine therapy (AI or fulvestrant) as the first-line treatment between 2018 and 2023. Of the included patients, 98% were women, with a median age of 62 years old (range between 53 and 70 years old). Endocrine therapy consisted in aromatase inhibitors in 84.9% of the patients and fulvestrant in 15.1% of the patients. Most patients were treated with the CDK 4/6i-and-AI combination in the first-line setting (103 patients), while 22 patients had previously been treated with endocrine therapy. 

In our cohort, the median OS for the entire group was 39 months, with survival data NA yet in the Palbo and AI group and 50.8 (95%CI, 32.9-NR) months in patients treated with Palbo and fulvestrant. The median TTD was higher in the Palbo and AI vs. Palbo and fulvestrant group (NA vs. 25 months). 

Adherence to Palbo in our cohort was remarkably high, with an average adherence rate of 0.9 ± 0.1. Interestingly, despite this high adherence, we observed no significant correlation between adherence and OS or TTD, suggesting that other factors, such as patient characteristics and tumor biology, might play a more prominent role in determining survival. The high Palbo adherence in our study was consistent with findings from other studies using the same measurement method (proportion of days covered) [17]. Our findings represent the first results regarding the association between Palbo adherence and OS in the real-world practice setting. We did not identify any risk regarding non-adherence on low OS or early discontinuation. It is difficult to understand in an analysis with claims data what the drivers of low Palbo adherence are, and this requires designing programs that monitor adherence and improving patient outcomes.

Our study reports data on the first experience of treating HR-positive/HER2-negative metastatic breast cancer with Palbo in Romania since it was firstly approved in 2018. As a real-world study, the population included is more heterogenous than the clinical trial. P-REALITY study published on the efficacy of Palbo in real-world setting, showing that the 24-month OS rate was 78.3%, and 65.8% remained alive at 36 months [18]. Another European real-world study [19] showed that patients treated with Palbo and AI had a median 24-month OS of 90.1%. The 24-month rate observed in our study is consistent with these previously reported data (78.2%), with a median OS not reached during follow-up. The median OS in the prospective POLARIS study [20] was 50.8 months in patients treated with Palbo and AI in combination, supporting the updated OS results from PALOMA-2 (median OS of 53.9 months, 95%CI: 49.8–60.8, after a follow-up of 90 months) [21], suggesting that a longer follow-up period may be necessary in order to assess OS in patients treated with AI in combination with a CDK4/6i.

In our cohort, the use of Palbo with fulvestrant showed a median TTD of 25 months and a median OS of 50.8 months. Although the number of patients treated with this combination was limited, existing data support the efficacy of fulvestrant in this setting. Moreover, the observed TTD is consistent with findings from other real-world studies (7.8–13.1 months) [22,23,24,25,26].

Additionally, a systematic review focusing on older patients with HR+/HER2− metastatic breast cancer found that Palbo, when combined with endocrine therapy, was effective and well tolerated, preserving quality of life regardless of age [27]. Moreover, although dose modifications, dose reductions, and treatment discontinuation rates for Palbo were higher in older patients compared to younger patients, these differences did not affect efficacy outcomes [27].

Although our study included a predominantly female population (98%), this aligns with the epidemiology of breast cancer, where only 1–2% of cases occur in males [28]. While our cohort included male patients, their numbers were insufficient for a dedicated subgroup analysis. This limitation reflects the rarity of male breast cancer and the subsequent challenge of conducting large-scale studies in this population.

However, other studies have explored the use of Palbo in male patients with HR+/HER2− advanced breast cancer. For example, a pooled analysis of male patients treated with Palbo in combination with endocrine therapy demonstrated outcomes consistent with those observed in female patients, supporting the efficacy and safety of this regimen across genders [29,30].

A recent analysis from the POLARIS study by Blum et al. (2024) provides valuable real-world insights into outcomes for male patients with HR+/HER2− advanced breast cancer [31]. This study demonstrated that male patients receiving Palbo in combination with endocrine therapy achieved outcomes comparable to those observed in female patients, including PFS and OS. These findings underscore the efficacy and safety of Palbo in male breast cancer, reinforcing its role as a treatment option for this rare but clinically significant subgroup.

Although we did not collect direct quality-of-life (QoL) data, high adherence often reflects favorable tolerability and patient satisfaction with treatment. Supporting this, the POLARIS study assessed QoL in patients with HR+/HER2− advanced breast cancer receiving Palbo plus endocrine therapy, using the EORTC QLQ-C30 questionnaire. The results showed that patients maintained their QoL over at least 18 months of treatment, with stable scores in global health status and functional scales, and no significant increase in symptom burden [32]. These findings suggest that the high adherence observed in our cohort may correlate with sustained QoL during Palbo treatment, as reported in other studies [33,34]. 

While the Romanian healthcare system provides full reimbursement for CDK4/6i for advanced breast cancer patients, access to care could still have been influenced by the COVID-19 pandemic. However, our analysis of patients with breast cancer initiating CDK4/6i therapy between 2018 and 2021 demonstrated that the COVID-19 pandemic did not impact treatment adherence. Previous research has indicated that the COVID-19 pandemic did not pose significant logistical challenges regarding the availability of oncologists in Romania, which might otherwise have influenced follow-up care [35]. Similarly, Navarro-Sabate et al. reported that the first wave of COVID-19 had only a modest impact on adherence to endocrine therapy for breast cancer in Spain [36].

Our study had several limitations. Firstly, the cohort was limited to the Dolj County, and the patient clinical characteristics could be different in other parts of Romania. Secondly, limited data access can be a source of bias in terms of interpreting the data, as we were not able to access all patient data in terms of dose reductions and tolerance, patient history, comorbidities, and disease severity. Moreover, routine quality of life data based on questionnaires are not collected in clinical practice in Romania. We suggest that future research focuses on expanding the cohort size and pooling data from multiple institutions or regions to strengthen the statistical significance of subgroup analyses. Nonetheless, we believe our findings serve as an important foundation for understanding treatment patterns and outcomes in this population. 

Although our study focuses on Palbociclib with hormonal therapy, surgery may be relevant in select cases of de novo metastatic disease. Recent research [37] found no significant survival benefit from surgery across most subgroups, but noted a slight trend toward better progression-free outcomes in triple-positive patients. These results highlight the importance of systemic therapies and the need for further prospective trials to clarify surgery’s role in this setting.

Despite these limitations, this study provides important information on the real-world efficacy of Palbo in Romanian patients with HR-positive/HER2-negative metastatic breast cancer, supporting the pre-existing trial and real-world data. 

## 5. Conclusions

In conclusion, our study contributes to the growing body of real-world evidence supporting the efficacy and safety of Palbo combined with endocrine therapy in HR-positive, HER2-negative breast cancer. While both AI and fulvestrant remain viable options, the lack of significant differences in survival between these combinations suggests that treatment choice can be tailored to individual patient needs. The findings from both our study and the broader global data underscore the importance of early CDK4/6 inhibitor intervention to optimize patient outcomes. Moving forward, further studies focusing on understanding the predictors of long-term adherence and outcomes in diverse populations are crucial to improving breast cancer management worldwide.

## Figures and Tables

**Figure 1 cancers-16-04161-f001:**
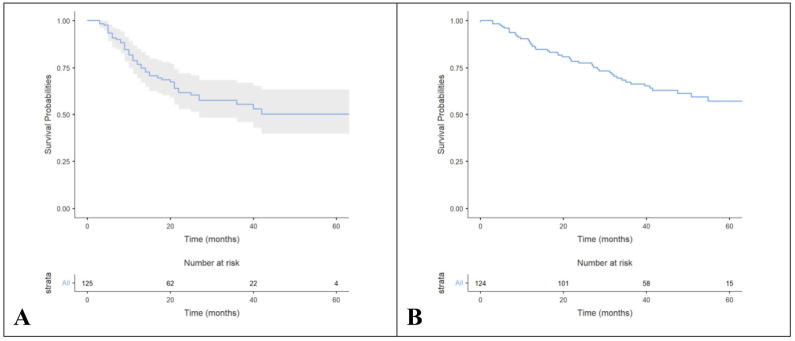
Kaplan–Meier plots for time to treatment discontinuation (TTD) and overall survival (OS). (**A**) TTD in all population; (**B**) OS in all population. Shaded area represents the confidence interval 95%.

**Figure 2 cancers-16-04161-f002:**
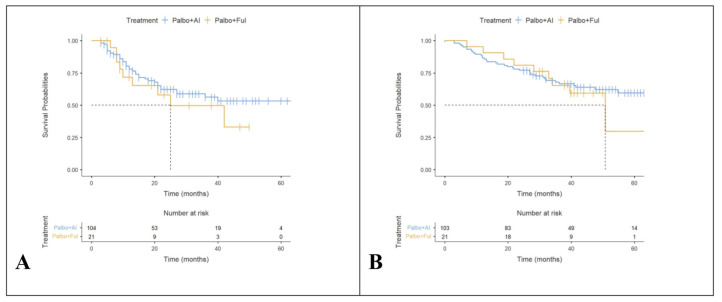
Kaplan–Meier plots for time to treatment discontinuation (TTD) and overall survival (OS). (**A**) TTD by endocrine treatment line; (**B**) OS by endocrine treatment line. Note: ‘+’ points along the curves represent censored observations.

**Figure 3 cancers-16-04161-f003:**
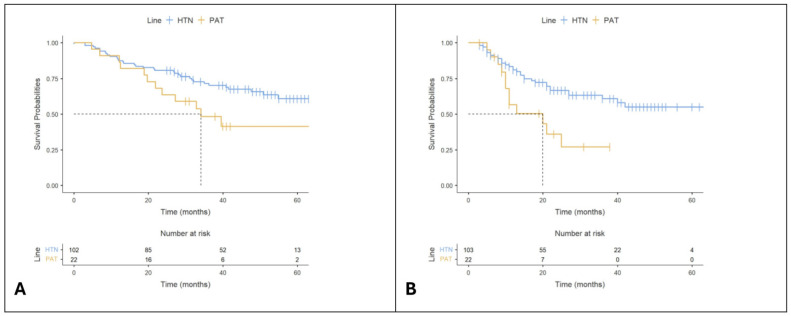
Kaplan–Meier plots for time to treatment discontinuation (TTD) and overall survival (OS). (**A**) TTD; (**B**) OS. Statistical significance was analyzed between the two types of treatment. Note: ‘+’ points along the curves represent censored observations. HTN, Hormone Therapy-Naïve; PAT, Prior Adjuvant Therapy.

**Figure 4 cancers-16-04161-f004:**
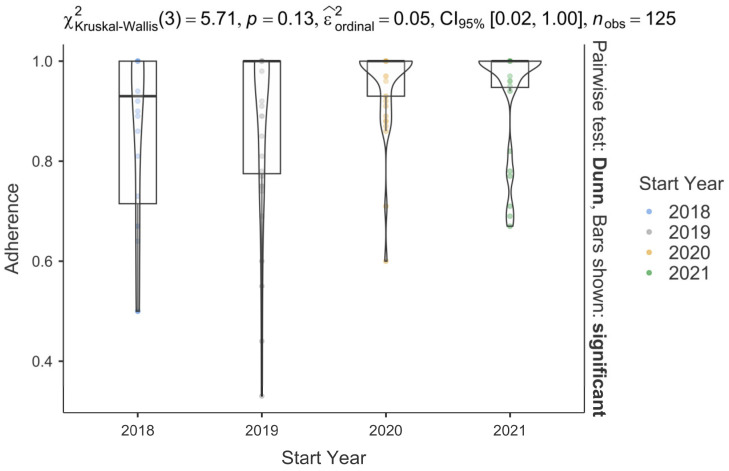
Adherence assessment among the starting years of Palbo treatment.

**Figure 5 cancers-16-04161-f005:**
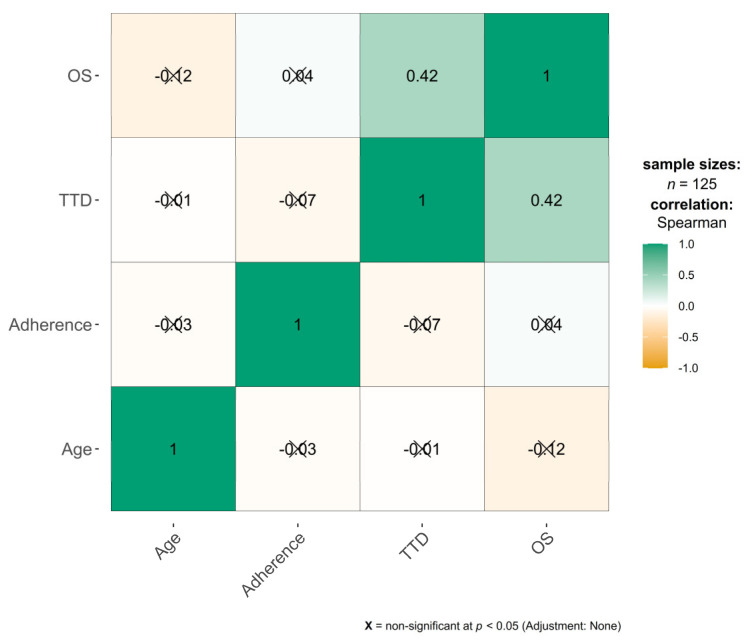
Heatmap of correlation coefficients between OS, TTD, adherence to Palbo, and age. Green corresponds to positive correlation (ρ = 1) and orange corresponds to negative correlation (ρ = −1).

**Table 1 cancers-16-04161-t001:** 1-, 2-, 3-, and 5-year TTD summary among the two types of combination treatment.

Treatment	12-Month TTD Rate	24-Month TTD Rate	36-Month TTD Rate	60-Month TTD Rate
Palbo + AI	77.8% [95%CI, 69.7–86.7%]	62.3% [95%CI, 52.7–73.16%]	56.3% [95%CI, 45.9–69.0%]	53.3% [95%CI, 42.4–67.1%]
Palbo + Fulvestrant	71.8% [95%CI, 53.6–96.2%]	58.0% [95%CI, 38.2–88.0%]	49.7% [95%CI, 29.7–83.2%]	-

**Table 2 cancers-16-04161-t002:** 1-, 2-, 3-, and 5-year OS summary among the two types of combination treatment.

Treatment	12-Month OS Rate	24-Month OS Rate	36-Month OS Rate	60-Month OS Rate
Palbo + AI	86.5% [95%CI, 80.2–93.4%]	76.9% [95%CI, 69.2–85.5%]	67.9% [95%CI, 59.2–77.8%]	59.6% [95%CI, 49.6–71.5%]
Palbo + Fulvestrant	95.2% [95%CI, 86.6–100.0%]	81.0% [95%CI, 65.8–99.6%]	65.3% [95%CI, 47.4–90.0%]	29.7% [95%CI, 7.1–100%]

**Table 3 cancers-16-04161-t003:** 1-, 2-, 3-, and 5-year TTD summary among the two lines of combination treatment.

Treatment	12-Month TTD Rate	24-Month TTD Rate	36-Month TTD Rate	60-Month TTD Rate
HTN	82.6% [95%CI, 75.1–90.7%]	68% [95%CI, 58.6–78.9%]	62.1% [95%CI, 51.7–74.5%]	56.1% [95%CI, 44.6–70.6%]
PAT	56.7% [95%CI, 38–84.7%]	36% [95%CI, 18.6–69.6%]	27% [95%CI, 11.3–64.4%]	-

HTN, Hormone Therapy-Naïve; PAT, Prior adjuvant therapy.

**Table 4 cancers-16-04161-t004:** 1-, 2-, 3-, and 5-year OS summary among the two lines of combination treatment.

Treatment	12-Month OS Rate	24-Month OS Rate	36-Month OS Rate	60-Month OS Rate
HTN	88.9% [95%CI, 82.9–95.3%]	81.8% [95%CI, 74.6–89.8%]	72.4% [95%CI, 63.8–82.1%]	61.3% [95%CI, 50.7–74.1%]
PAT	90% [95%CI, 77.8–100%]	60% [95%CI, 42–85.8%]	42.8% [95%CI, 25.2–72.6%]	34.2% [95%CI, 17.2–68%]

**Table 5 cancers-16-04161-t005:** Multivariable analysis assessing the effect of age, gender, or adherence on overall survival.

Factors	HR (Univariable) 95%CI, *p*-Value	HR (Multivariable) 95%CI, *p*-Value
Gender (female vs. male)	0.80 (0.11–5.82), *p* = 0.824	0.73 (0.1–5.38), *p* = 0.760
Treatment (Palbo + fulvestrant vs. Palbo + AI)	1.19 (0.57–2.48), *p* = 0.638	1.16 (0.55–2.42), *p* = 0.697
Age	1.00 (0.98–1.03), *p* = 0.686	1.01 (0.98–1.03), *p* = 0.641
Adherence	0.33 (0.05–2.39), *p* = 0.272	0.32 (0.04–2.36), *p* = 0.265

**Table 6 cancers-16-04161-t006:** Multivariable analysis assessing the effect of age, gender, or adherence on time to treatment discontinuation.

Factors	HR (Univariable) 95%CI, *p*-Value	HR (Multivariable) 95%CI, *p*-Value
Gender (female vs. male)	0.77 (0.11–5.59), *p* = 0.795	0.73 (0.1–5.32), *p* = 0.752
Treatment (Palbo + fulvestrant vs. Palbo + AI)	1.35 (0.65–2.8), *p* = 420	1.32 (0.63–2.75), *p* = 0.464
Age	1.00 (0.98–1.03), *p* = 0.907	1.00 (0.98–1.02), *p* = 0.927
Adherence	0.38 (0.05–2.79), *p* = 0.344	0.41 (0.06–2.98), *p* = 0.379

## Data Availability

The data are available upon request from the corresponding author.

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
