# Peer review of "Real-World Efficacy and Adherence to Palbociclib in HR-Positive, HER2-Negative Advanced Breast Cancer: Insights from a Romanian Cohort"

_cancers, 2024, doi:10.3390/cancers16244161_

Round 1

Reviewer 1 Report (Previous Reviewer 1)

Comments and Suggestions for Authors

This manuscript investigates the real-world outcomes of palbociclib (Palbo) combined with endocrine therapy in HR+/HER2− metastatic breast cancer in a Romanian cohort. The study analyzes adherence, overall survival (OS), and time to treatment discontinuation (TTD) using retrospective data. The authors report high adherence (mean adherence of 0.91 ± 0.1) and median OS of 39 months for the entire cohort, with 50.8 months for patients treated with Palbo and fulvestrant. TTD was longer for Palbo with AI compared to fulvestrant. The findings align with international studies, highlighting Palbo's real-world efficacy.

The manuscript is well-structured and provides valuable insights, especially in an underrepresented population. However, several critical issues should be addressed:

Data Limitations: The reliance on health insurance claims data restricts the scope of analysis. Missing information on reasons for treatment discontinuation, dose modifications, comorbidities, and disease progression limits the conclusions on adherence and efficacy.

Adherence Analysis: Despite high adherence, the lack of correlation with OS or TTD raises concerns about unmeasured confounders such as tumor biology or patient-specific factors. The authors should discuss possible explanations and limitations in depth.

Male Subgroup: The inclusion of male patients is commendable, but the sample size is insufficient for robust subgroup analysis. While consistent with male breast cancer epidemiology, the manuscript could better contextualize the limitations of such analysis.

Quality of Life (QoL): The absence of QoL data, which could substantiate adherence findings, is a notable limitation. The authors could reference studies that link QoL with adherence to improve interpretation.

Statistical Adjustments: The lack of multivariable analyses adjusting for baseline characteristics in key comparisons (e.g., AI vs. fulvestrant) weakens the generalizability. A sensitivity analysis could enhance robustness.

Real-World Setting: While the focus on a Romanian cohort is novel, the authors should elaborate on how healthcare disparities, regional treatment access, or socio-economic factors may influence outcomes.

Treatment Options: Some alternative treatment options such as surgery should be discussed by the authors to improve their manuscript. Please cite PMID: 36551722 which reported encouraging oncological results in this setting of disease.

Author Response

Data Limitations: The reliance on health insurance claims data restricts the scope of analysis. Missing information on reasons for treatment discontinuation, dose modifications, comorbidities, and disease progression limits the conclusions on adherence and efficacy.

Response: Thank you for pointing this out. The scope of our analysis was to evaluate the time to treatment discontinuation (TTD), and overall survival (OS) for the first time in Romania for a CDK4/6i, Palbociclib, in patients with HR-Positive, HER2-Negative Advanced Breast Cancer. Missing information on reasons for treatment discontinuation, dose modifications, comorbidities, and disease progression limits our conclusions related to the factors influencing OS and TTD levels. This was mentioned in limitations paragraph in the revised manuscript (lines 357-360, 396-405).

Adherence Analysis: Despite high adherence, the lack of correlation with OS or TTD raises concerns about unmeasured confounders such as tumor biology or patient-specific factors. The authors should discuss possible explanations and limitations in depth.

Response: Tumour biology is not a confounder. Concerning the molecular subtype, as per other clinical trials that evaluated CDK4/6i as treatment for metastatic breast cancer patients, only luminal subtypes were included, with HR positive and Her2 negative patients. This discuss can be found in the revised manuscript (lines 326-336).

Male Subgroup: The inclusion of male patients is commendable, but the sample size is insufficient for robust subgroup analysis. While consistent with male breast cancer epidemiology, the manuscript could better contextualize the limitations of such analysis.

Response: Data regarding male breast cancer are scarce and we have considered important to present them even if the sample size is small. The discussion can be found in the revised manuscript (lines 361-377).

Quality of Life (QoL): The absence of QoL data, which could substantiate adherence findings, is a notable limitation. The authors could reference studies that link QoL with adherence to improve interpretation.

Response: We agree the evaluation of health-related quality of life is important in routine clinical practice, but the Romanian database does not save these assessments and they were missing in the current study. We have discussed this limitation in our manuscript (lines 378-386).

Statistical Adjustments: The lack of multivariable analyses adjusting for baseline characteristics in key comparisons (e.g., AI vs. fulvestrant) weakens the generalizability. A sensitivity analysis could enhance robustness.

Response: We have already performed the multivariable analyses as you requested first. We have accordingly modified the multivariable analyses taking into account the comparison between AI and fulvestrant, This change can be found in the revised manuscript (lines 255-264).

Real-World Setting: While the focus on a Romanian cohort is novel, the authors should elaborate on how healthcare disparities, regional treatment access, or socio-economic factors may influence outcomes.

Response: Romania's healthcare system provides full reimbursement for CDK4/6 inhibitors for advanced breast cancer patients, effectively reducing financial barriers to treatment access from the patient's perspective. To assess the potential impact of the COVID-19 pandemic on adherence to breast cancer treatment, we compared our results across the years 2018, 2019, 2020, and 2021, and no differences were found. Navarro-Sabaté et al. reported that the first wave of the COVID-19 pandemic had a modest impact on adherence to endocrine therapy for breast cancer in Spain. However, no other studies were identified that specifically address the influence of socio-economic disparities, patient education, awareness, or cultural beliefs on treatment adherence and outcomes in Romania. Furthermore, no regional variations in the implementation of guidelines or clinical practices that could potentially affect outcomes were observed in other research. These changes are reflected in the revised manuscript (lines 387-395).

Treatment Options: Some alternative treatment options such as surgery should be discussed by the authors to improve their manuscript. Please cite PMID: 36551722 which reported encouraging oncological results in this setting of disease.

Response: Although our study focuses on Palbociclib with hormonal therapy, surgery may be relevant in select cases of de novo metastatic disease. Recent research ([35], PMID: 36551722) found no significant survival benefit from surgery across most subgroups but noted a slight trend toward better progression-free outcomes in triple-positive patients. These results highlight the importance of systemic therapies and the need for further prospective trials to clarify surgery's role in this setting. This change can be found in the revised manuscript (lines 406-411).

Reviewer 2 Report (Previous Reviewer 2)

Comments and Suggestions for Authors

The authors made significant revisions based on the reviewer's comments; the manuscript can be acceptable for publication.

Author Response

Dear Reviewer,
Thank you for your time and effort in reviewing our manuscript. We sincerely appreciate the detailed and thoughtful feedback provided, which has been invaluable in improving the quality and clarity of our work.

Reviewer 3 Report (Previous Reviewer 3)

Comments and Suggestions for Authors

The authors have addressed the concerns I raised. It can be published in the present form.

Author Response

Dear Reviewer,
Thank you for your time and effort in reviewing our manuscript. We sincerely appreciate the detailed and thoughtful feedback provided, which has been invaluable in improving the quality and clarity of our work.

Reviewer 4 Report (Previous Reviewer 4)

Comments and Suggestions for Authors

Thank you for taking my comments into consideration and congratulations for this improved manuscript.

Author Response

Dear Reviewer,
Thank you for your time and effort in reviewing our manuscript. We sincerely appreciate the detailed and thoughtful feedback provided, which has been invaluable in improving the quality and clarity of our work.

Round 2

Reviewer 1 Report (Previous Reviewer 1)

Comments and Suggestions for Authors

The manuscript can be accepted in the present form

This manuscript is a resubmission of an earlier submission. The following is a list of the peer review reports and author responses from that submission.

Round 1

Reviewer 1 Report

Comments and Suggestions for Authors

This manuscript represents real-world data on efficacy and adherence to Palbociclib in combination with endocrine therapy in 119 HR+/HER2- breast cancer patients from Romania. The authors report a median overall survival of 39 months, with a very high adherence rate to Palbociclib that is not associated significantly with any outcome.

There are many limitations to this study that really weaken the conclusions.

First, the sample size is small, 119 patients, and this becomes even more insignificant when sub-divided by treatments for derived conclusions that are meaningful or reaching statistical significance.

Also, the follow-up period is relatively short in assessing OS in metastatic breast cancer. The absence of detailed demographic data such as cancer stage and molecular subtype restricts their interpretability and generalizability to the larger population. Therefore, further exclusion of the patients treated less than five months and also having missing mortality data narrows the cohort, which increases the potential selection bias.

Another major flaw is that no adjustments for confounding factors have been carried out, and the analysis presented is unadjusted.

Future studies should overcome the deficiencies of the present one by including larger and more diverse patient cohorts with comprehensive demographic and clinical data and longer follow-up to achieve stronger results.

Reviewer 2 Report

Comments and Suggestions for Authors

The article "Real-World Efficacy and Adherence to Palbociclib in HR-Positive, HER2-Negative Advanced Breast Cancer: Insights from a Romanian Cohort" has been reviewed. This study aimed to evaluate the effectiveness, safety, and adherence to Palbociclib in combination with aromatase inhibitor (AI) or fulvestrant in a real-world group of HR+/HER2- breast cancer patients from Romania. The researchers conducted a retrospective analysis of reimbursed Palbociclib prescriptions using data from the electronic database of the Romanian Health Insurance House, Dolj County, for disease code 124, covering the period from 2018 to 2023.

The primary outcome assessed was time to treatment discontinuation (TTD), with secondary outcomes including overall survival (OS) and Palbociclib adherence (measured by medication possession ratio). A total of 119 patients were identified, with a median age of 62 years (IQR, 53-70), and 98% were female. Two treatment combinations were observed: Palbociclib + Aromatase Inhibitor (AI) in 101 patients (84.9%) and Palbociclib + Fulvestrant in 18 patients (15.1%).

The median TTD for the entire cohort was 20 months (95% CI, 20.2-25.9 months). In patients treated with Palbociclib + AI, the median TTD was not available/reached [NA] (95% CI, 36.0-NA months). When the treatment was Palbociclib + Fulvestrant, the median was 50.8 (95% CI, 32.9-NA) months. No significant differences were found between the four types of treatments in terms of overall survival (log-rank p = 0.249). The 24- and 36-month OS rates were higher for Palbociclib combined with AI than combined with Fulvestrant: 78.2% [95% CI, 70.6%-86.7%] vs. 77.8% [95% CI, 60.8%-99.6%], and 69.0% [95% CI, 60.3%-79.0%] vs. 59.1% [95% CI, 39.6%-88.1%], respectively.

The mean adherence in our study was 0.9±0.1. We found no correlation between adherence to Palbociclib and OS (Spearman’s rho = 0.05, p = 0.593). While both AI and fulvestrant remain viable options, the lack of significant differences in survival between these combinations suggests that treatment choice can be tailored to individual patient needs. The article needs to be rewritten to attract readers' attention, it needs major revision before consideration.

1.      Please revise the introduction to focus on cancer statistics and then the drug combination of Palbociclib and aromatase inhibitor. Additionally, discuss breast cancer management.

2.      Enhance the resolution of figures (1-4) and ensure that the text in the figure images is legible.

3.      The current number of references is inadequate; it is recommended to include recent studies to support the points made in the manuscript.

4.      It is suggested to reduce the similarity index to below 20%, as the current manuscript exceeds 29% in similarity index.

5.      Furthermore, the authors are encouraged to include the strengths and weaknesses of the study.

Comments on the Quality of English Language

 Moderate editing of English language required.

Reviewer 3 Report

Comments and Suggestions for Authors

This manuscript presents a retrospective study of 191 ER+Her2- metastatic breast cancer patients treated with palbociclib (PAL) in combination with either aromatase inhibitors (AI) or fulvestrant (FUL). Of the patients, 101 were treated with PAL+AI and 18 with PAL+FUL. The study compared these two treatment groups for time to treatment discontinuation (TTD) and overall survival (OS). Statistical analysis revealed no significant differences in TTD or OS between the two groups.

As a real-world efficacy and adherence analysis, this study adds value, particularly given the limited data comparing PAL+FUL and PAL+AI in treating metastatic ER+Her2- patients. Overall, the manuscript is well-written with robust statistical analysis, but there are several issues that require clarification:

  1. Study Population Clarification: In section 3.1, it should be explicitly stated that all patients had metastatic breast cancer and were receiving PAL plus endocrine therapy as first-line treatment.
  2. Line 1 vs. Line 2 Therapy: In Figures 3, Tables 3, and 4, the distinction between "line 1" and "line 2" therapy is unclear. If "line 1" refers to first-line therapy and "line 2" to second-line therapy, the authors should explicitly define this. Moreover, the discussion states that all 119 patients received therapy as first-line treatment, which contradicts the second line therapy and needs clarification.
  3. Analysis by Starting Year: In Figure 4 and Table 6, the rationale for analyzing starting year as a factor is unclear. The manuscript does not mention any differences in age, gender, disease grades, or stages among these groups, making the analysis results less reliable. Clarification is needed.
  4. Multivariate Analysis Request: In Figure 5, instead of using correlation analysis, the authors should conduct a multivariate analysis to determine whether factors such as age, gender, and adherence influence TTD and OS.

Comments on the Quality of English Language

Needs some editing on the writings. 

Reviewer 4 Report

Comments and Suggestions for Authors

I wish to congratulate the authors for this well-written paper.

The introduction should also mention the key trial results with the other 2 CDK4/6 inhibitors currently used in the clinic and put their use into perspective. When discussing palbo, it is in my opinion wrong to state that it "particularly" halts cell cycle progression.

Please specify what you mean by "disease code 124".

Excluding patients who received palbo for less than 5 months excludes the worst cases, thus improving the outcomes. This should be explained. If the explanation is satisfactory, it should be discussed. If not, the analyses should be repeated after inclusion of theses patients.

There are 2 3.2.4 sections. The first one is very repetitive and should be simplified.

It is not clear in which line each combination is used in the studied patient population. This should be better described.

Comments on the Quality of English Language

Minor editing is required. In particular, replace "pablo" with "palbo".